# Understanding Efficiency: Quantization, Batching, and Serving Strategies in LLM Energy Use

## Abstract

Large Language Models (LLMs) are increasingly deployed in production, contributing towards shifting the burden in terms of computational resources and energy demands from training to inference. While prior work has examined the energy cost of inference per prompt or per token, we highlight how **system-level design choices** - such as numerical precision, batching strategy, and request scheduling - can lead to orders-of-magnitude differences in energy consumption for the same model. We perform a detailed empirical study of LLM inference energy and latency on NVIDIA H100 GPUs, analyzing the impact of quantization, batch size, and serving configuration (e.g., with Hugging Face's Text Generation Inference server). Our results reveal that lower-precision formats only yield energy gains in compute-bound regimes; that batching improves energy efficiency, especially in memory-bound phases like decoding; and that structured request timing (arrival shaping) can reduce per-request energy by up to $100\times$. We argue that sustainable LLM deployment depends not only on model internals, but also on the orchestration of the serving stack. Our findings motivate phase-aware energy profiling and system-level optimizations for greener AI services.

## 1 Introduction

As large language models (LLMs) transition from research prototypes to real-world services, the energy consumed during inference has become a growing concern (Wu et al., 2021; Luccioni et al., 2024b; Everman et al., 2023). While training once dominated the environmental cost of AI (Strubell et al., 2019; Patterson et al., 2021), the widespread deployment of LLMs in user-facing applications - chatbots, assistants, code generators - has shifted the spotlight toward inference efficiency. Studies have called for improved energy reporting (Henderson et al., 2020; Luccioni et al., 2024b), quantified the energy per query (Samsi et al., 2023), and highlighted the role of verbosity (Poddar et al., 2025; Gao et al., 2024; Jin et al., 2025) and prompt length (Wilkins et al., 2024a) in power consumption. However, inference efficiency remains underexplored in dynamic serving conditions (Fernandez et al., 2025).

In this paper, we explore these systemic factors in detail. Building on prior work measuring the energy footprint of specific prompts or models (Luccioni et al., 2024b; Samsi et al., 2023; Everman et al., 2023), we shift focus from the "what" of user inputs to the "how" of inference delivery. Through extensive experiments on NVIDIA H100 GPUs, we analyze how quantization (Dettmers et al., 2022; 2023), batching (Liu et al., 2024; Agrawal et al., 2023), and dynamic serving (Face, 2022; Dao, 2023; Yu et al., 2022) affect both latency and energy consumption, phase by phase.

Our key contributions are:

- A detailed evaluation of five numerical precisions across multiple models, revealing when quantization helps - and when it backfires - in memory-bound regimes.

- A systematic study of batch size effects, including normalized energy per useful token and the trade-offs introduced by input/output padding.

- An empirical benchmark of Hugging Face's Text Generation Inference (TGI) server under varying traffic patterns, showing that arrival shaping can drastically improve batching quality and reduce per-request energy up to $100\times$.
- A practical synthesis of system-level insights for building energy-efficient LLM inference pipelines.

All scripts, configurations, and measurement tools used in this study are available open-source. results.

Our findings suggest that inference efficiency is not just a function of model internals, but of the entire serving stack - from GPU kernels to traffic scheduling. Optimizing this stack can yield substantial sustainability gains, even without modifying the underlying model.

## 2 EXPERIMENTAL SETUP

We benchmarked a selection of some of the most downloaded instruction-tuned open-source LLMs on Hugging Face as of July 2025, focusing on standard model sizes in the range of a few billion parameters. Our benchmark includes:

- **Qwen 2.5**: 0.5B, 1.5B, 3B, 7B, 14B
- **Mistral-7B-Instruct-v0.3**
- **LLaMA 3.1–8B-Instruct**

Each model was evaluated under five numerical formats:

- `float32`, `bfloat16`, `float16` (native support via PyTorch)
- `int8`, `int4` using `bitsandbytes` (2020) quantization (via the LLM.int8() and LLM.int4() formats).

For `int8` and `int4`, we applied post-training quantization using `bitsandbytes`, which compresses the feed-forward and attention projection weights using vector-wise quantization. For `int8`, LLM.int8 performs 8-bit matrix multiplications with outlier-aware mixed precision, isolating rows or columns with large activation features and computing them in 16-bit to preserve accuracy Dettmers et al. (2022). For `int4`, weights are packed two per byte and stored in a NormalFloat4 (NF4) format; custom CUDA kernels perform on-the-fly dequantization before matmuls (Dettmers et al., 2023).

All models were loaded and executed using the `Transformers` library (Wolf et al., 2020), which by default leverages optimized kernels such as FlashAttention and fused operations provided by recent PyTorch releases.

All runs were conducted on a dedicated NVIDIA H100 SXM GPU (80GB) and 8 AMD EPYC 7R13 CPU cores, with no co-scheduled jobs. GPU and CPU energy were measured using the `CodeCarbon` library (Courty et al., 2024), which leverages NVML and pyRAPL for real-time energy monitoring, while RAM energy was estimated via a CodeCarbon heuristic [1] based on CPU count and usage duration. Latency was recorded at the CUDA kernel level.

Each request was preceded by a warmup phase of 5 iterations to stabilize memory and kernel behavior. For each configuration, we repeated the same request 10 times and report the average energy and latency to reduce variability.

We reused the a subset of the dataset proposed by (Anonymous & Anonymous, 2025) (under review), which studies the energy impact of polite interactions with LLMs. This dataset provides a controlled and reproducible input distribution while preserving real-world relevance. Specifically, we used 10,000 polite prompts (ending in "thank you") sampled from a custom subset of the UltraChat-200k dataset (Ding et al., 2023), available at `ultrachat 10k`. Prompts ranged from 200 to 4000 tokens, and outputs were relatively short - typically between 10 and 300 tokens - due to the nature of the dataset, which consists of chats between a human user and an LLM. Prompts were adapted to match the input format expected by each model.

---

[1] https://mlco2.github.io/codecarbon/methodology.html#ram

To analyze energy and latency independently for *prefill* and *decode*, we split the inference into two steps:

- **Prefill**: Forward pass over the full prompt (with generation stopped at the first token).
- **Decode**: Autoregressive generation of the remaining tokens, attending to cached context.

The full `generate` phase corresponds to the sum of *prefill* and *decode*. In practice, we isolate *prefill* by generating a single token, and obtain *decode* as the difference between the full generation and the prefill run. This decomposition enables us to capture the distinct compute regimes that characterize each phase:

The *prefill* phase is not uniformly compute-bound: for very small input sizes, most operations are memory-bound due to limited arithmetic intensity. (Memory-bound operations are limited by data movement rather than computation; although memory and compute can be executed asynchronously on GPUs, one or the other often becomes the bottleneck, depending on the workload.) As the input length ($s$) increases, compute-heavy operations - such as feedforward layers and QKV (query, key, and value) projections - begin to dominate, especially in large models with wider hidden dimensions. Compute-bound operations, by contrast, are limited by the rate at which arithmetic can be performed. The transition point from memory-bound to compute-bound depends primarily on the model's hidden size, with larger models entering the compute-bound regime earlier. Increasing the batch size also accelerates this transition by increasing the FLOP-to-memory ratio.

In contrast, the *decode* phase remains fully memory-bound for small batch size, regardless of model size. This is due to the autoregressive nature of generation: each token is produced sequentially and involves computing attention over cached prompt representations at each decoding step, leading to small, fragmented memory operations. Only by increasing the batch size does the decode phase start to exhibit compute-bound characteristics.

**Idle time.** GPU utilization can be impacted by idle times between kernels. When the CPU thread issuing kernels is slower than the GPU execution, the GPU may stall despite its asynchronous capabilities - leading to gaps where no work is scheduled. This underutilization becomes more pronounced in workloads with small or irregular kernel launches.

## 3 IMPACT OF NUMERICAL PRECISION ON LATENCY AND ENERGY CONSUMPTION

As LLMs grow in size, the adoption of lower-precision numerical formats - such as `bfloat16`, `int8`, or `int4`-has become a widespread strategy to reduce memory footprint and enable inference for larger models on constrained hardware. While these formats can also improve throughput and hardware utilization, their actual benefits are often phase-dependent and not always straightforward. In this section, we dissect how numerical precision impacts both latency and energy consumption across the two main phases of inference: *prefill* and *decode*. We show that precision reduction yields significant gains primarily in compute-bound regimes, whereas in memory-bound settings, aggressive quantization may introduce dequantization overheads or bandwidth saturation that offset the expected improvements.

### 3.1 PREFILL PHASE: ACCELERATION WITHOUT PROPORTIONAL ENERGY SAVINGS

In the prefill phase, we observe up to **4× reduction in GPU energy** when switching from `float32` to lower-precision formats such as `float16`, `bfloat16`, or `int8` - particularly for larger models (e.g., LLaMA 8B or Qwen 14B) - see Figure 1a. These models are predominantly compute-bound at the input lengths seen in our dataset (typically $s_{mean} \approx 1200$), and benefit fully from the activation of **Tensor Cores**, which enable fused matrix multiplications with up to $15\times$ higher throughput.

Smaller models, in contrast (e.g., Qwen-0.5B and 1.5B), remain memory-bound across most of the prompt lengths we tested, as their hidden sizes are smaller and their compute intensity lower. As a result, they gain little to no advantage from Tensor Core acceleration. In some cases, we even observe a slight increase in energy consumption for `float16`/`bfloat16`, likely due to the activation of

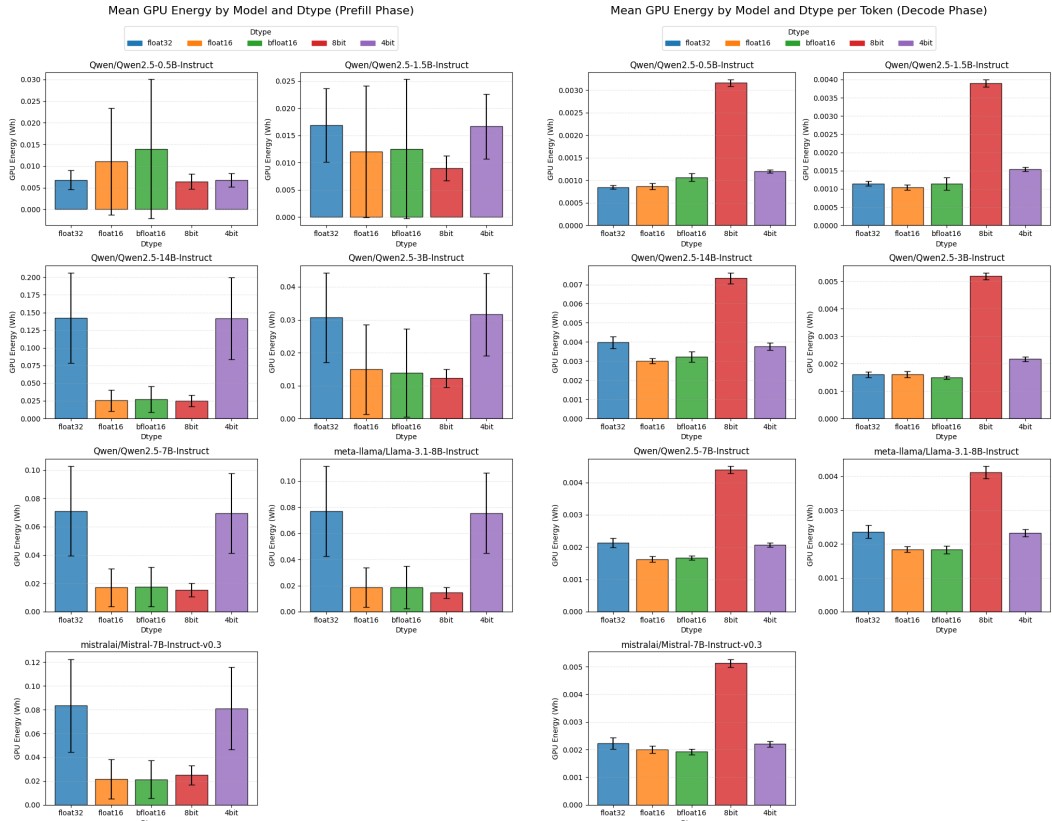

(a) Mean GPU energy consumption by model and dtype during the prefill phase.

(b) Mean GPU energy consumption per token by model and dtype during the decode phase.

Figure 1: Impact of model size and numerical precision (dtype) on GPU energy consumption during (a) prefill and (b) decode phases.

specialized compute kernels (Tensor Core paths) that add overhead without enough computations to amortize it (Figure 1a).

For quantized models (`int8` and `int4`), performance is further impacted by on-the-fly dequantization: during inference, weights stored in compressed integer formats are unpacked and converted to higher-precision tensors (typically `float16`/`bfloat16` or `float32`) before computation. This unpacking adds extra kernel launches and memory movement, which can partially negate the benefits of quantization, especially when the operations are memory-bound or irregular.

While latency does decrease significantly in many of these configurations - up to $10\times$ in large models (Figure 4) - the energy savings are smaller. This is due to a higher **average power consumption** when using Tensor Cores: they complete the computation faster, but at a higher instantaneous power draw. As a result, the time is shorter but the power is higher, limiting the total energy saved.

### 3.2 DECODE PHASE: QUANTIZATION PITFALLS IN MEMORY-BOUND REGIMES

In contrast to prefill, the *decode* phase is fully memory-bound for all model sizes and sequence lengths considered. Each generated token reuses cached activations (KV caching) and performs attention over the accumulated context, with little opportunity for parallel compute acceleration.

As a result, energy per generated token remains largely invariant across float32, float16, and bfloat16, with minor improvements (or slight degradations) in both energy (Figure 1b) and latency (Figure 5). This suggests that lower-precision Tensor Cores do not provide significant benefits in this memory-bound regime. *Theoretically*, in a bandwidth-limited regime, - and thus latency and energy per token - should scale inversely with the memory word size $b_w$: reducing from `float32` (32 bits) to

float16 (16 bits) or int8 (8 bits) should yield ideal 2× or 4× gains, respectively. However, such improvements are not observed in practice.

The reason lies in the energy profile of memory-bound workloads: while kernels may run slightly faster with lower precision, the GPU spends a disproportionate amount of time idle between kernel launches, waiting for synchronization, scheduling, or small fragmented memory operations. Since GPU idle power remains non-negligible - typically around 120 W even when no kernel is running - reducing kernel duration has little effect on total energy per token. The energy saved from faster compute is offset by the energy burned during idle time.

Quantized formats like int8 and int4 further exacerbate this issue: they introduce additional dequantization kernels that are small, memory-bound, and irregular, increasing the number of launches and stream fragmentation. As a result, we observe higher energy consumption with int8-often 2–3× more than float32-despite moving fewer bytes (Figure 1b).

Modern GPUs also transfer memory in fixed-width chunks (e.g., 32–64 bytes), so 4-bit formats do not reduce memory bandwidth proportionally. Combined with memory misalignment and suboptimal coalescing, this results in negligible or even negative energy gains from quantization in the decode phase. In fact, we find that int4 performs similarly to float32, reinforcing the notion that in memory-bound phases with high kernel fragmentation, reducing numerical precision is insufficient to meaningfully reduce energy use.

**In summary:** numerical precision reduction yields the most benefit in the *prefill* phase of large models, where compute dominates. In contrast, the *decode* phase remains memory-limited, and aggressive quantization (e.g., int8 or int4) may incur overheads that outweigh theoretical savings.

## 4 BATCH SIZE EFFECTS ON ENERGY EFFICIENCY

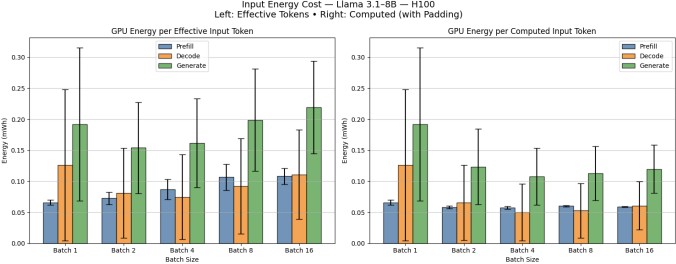
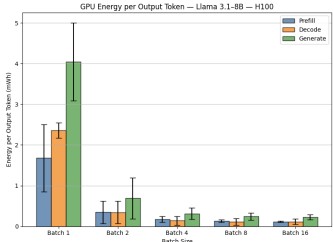

(a) GPU energy per **input token**. Left: Effective tokens (excluding padding); Right: Computed tokens (including padding).

(b) GPU energy per **output token** (effective = computed).

Figure 2: GPU energy consumption per token on LLaMA 3.1–8B. (a) Input-side energy depends on token type and padding; (b) Output-side energy remains consistent across requests.

Batching is one of the most effective levers for improving throughput and reducing per-request overhead in LLM inference. By processing multiple sequences in parallel, batching amortizes fixed costs such as memory transfers and kernel launch overheads. However, its impact on energy consumption depends on the inference phase (prefill vs decode), the compute regime (compute- vs memory-bound), and the presence of padding. In this section, we analyze how GPU energy scales with batch size for **LLaMA 3.1–8B (float32)**, using the transformers library in static batching mode.

We separate the analysis into two perspectives:

- Energy per *input token*, distinguishing between **effective** (excluding padding) and **computed** (including padding) tokens;

- Energy per *output token*, where effective = computed since completed sequences are dropped automatically.

**Input token normalization: trade-offs between padding and parallelism.** To understand how batch size affects different phases of inference, we first normalize energy by the number of *input tokens*.

On the left of Figure 2a, the energy per *effective input token* in the prefill phase increases steadily with batch size due to padding. As sequences are padded to match the longest one, the compute-bound prefill phase performs extra work on padded tokens, leading to inflated energy per effective token.

In the decode phase, we observe a U-shaped curve: batching improves memory reuse and reduces launch overheads, but larger batches increase the number of prompt tokens per sequence, which in turn increases the cost of each attention step. The optimal batch size for decode is reached at $b = 4$ in our setup. This U-shape carries over to the total generate phase, which sees minimal energy per effective input token at $b = 2$, a compromise between prefill waste and decode gains. At $b = 16$, energy per token increases by nearly 25% compared to this optimal point.

When energy is normalized by *computed input tokens* (right of Figure 2a), a different picture emerges. Prefill energy per token remains constant, as expected for a compute-bound workload where energy scales linearly with FLOPs. Decode energy per computed token decreases with batch size, but the gains plateau around $b = 4$ as the marginal benefits of parallelizing attention computations diminish for the sequence lengths considered. The generate phase follows the same trend, reaching about 65% of the energy per token observed at $b = 1$.

**Output token normalization: efficient batching across all phases.** In Figure 2b, we normalize energy by the number of *output tokens*. Here, all tokens are **effective** because `transformers` automatically drop completed sequences from the batch, avoiding padding overheads.

We observe consistent improvements across all phases. Energy per output token decreases rapidly with batch size and follows a roughly logarithmic trend. The memory-bound *decode* phase benefits the most: matrix multiplications over cached keys and values dominate its cost, and batching enables better amortization of memory transfers. Prefill, though compute-bound, also appears more efficient in this metric, since its fixed cost is shared across a larger number of generated tokens. Lastly, full generation shows the same trend, confirming that larger batches lead to longer kernels and reduced idle times, further improving energy efficiency.

**Conclusion.** Batching improves energy efficiency across all inference phases, but through different mechanisms. In the *prefill* phase, gains are limited by padding, which inflates compute without contributing useful work. In the *decode* phase, batching brings strong benefits up to $b = 4$, after which parallelism yields diminishing returns. Normalizing by output tokens confirms consistent efficiency gains due to reduced overheads and longer, better-amortized kernels. Overall, optimal batch size depends on the chosen normalization and reflects a trade-off between parallelism and padding waste, with efficiency gains driven by improved compute saturation.

## 5 ENERGY EFFICIENCY WITH TGI AND ARRIVAL SHAPING

To simulate production-like deployment scenarios, we ran LLaMA 3.1–8B and 70B in `bfloat16` using Hugging Face's `text-generation-inference` (TGI) server (v3.3.4). TGI enables continous batching and integrates multiple inference optimizations such as more kernel fusion. This section investigates how usage patterns and in particular, request arrival timing affect - batching quality and energy efficiency.

### 5.1 METHODOLOGY

We evaluated the per-request energy consumption under different inter-arrival patterns:

- **Random delays**: each request $i$ is sent at $t_i = i \cdot \Delta$, with $\Delta \sim \mathcal{U}(k, l)$.
- **Fixed intervals**: regular delays between requests (e.g., every 50ms, 300ms, or 500ms).

In both cases, we sent 10,000 generation requests to the TGI server. Energy consumption was tracked via `nvml` on the GPU host, and averaged over all requests.

## 5.2 LLaMA 8B: Arrival Shaping Unlocks Large Gains

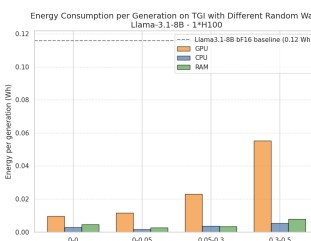 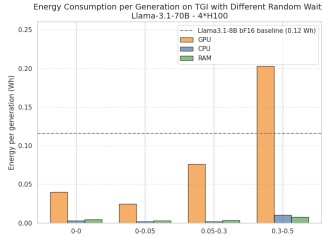 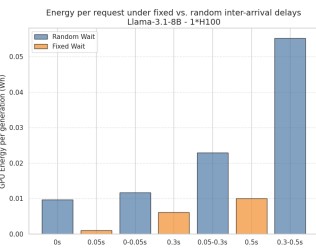

(a) LLaMA 8B: mean energy per request (GPU, CPU, RAM) under random arrival.

(b) LLaMA 70B: same setup as in (a), scaled up to larger model size.

(c) LLaMA 8B: energy per request under fixed vs. random inter-arrival delays.

Figure 3: Impact of inter-arrival delay and model size on energy per request. (a) and (b): Mean energy for LLaMA 8B and 70B under random delays. (c): Comparison of fixed vs. random delays at 8B scale.

For LLaMA 3.1–8B, switching from the standard `transformers` library (with sequential request handling) to Hugging Face's `text-generation-inference` (TGI) server (with burst-mode batching) reduces the mean energy per request from $1.2 \times 10^{-1}$ **Wh** to $9.6 \times 10^{-3}$ **Wh**. This $12.5\times$ improvement highlights the impact of continuous batching and backend optimizations in TGI (Figure 3a).

Further improvements are possible with fixed inter-arrival delays. As shown in Figure 3c, using a constant spacing of 500 ms reduces energy to as low as $1.1 \times 10^{-3}$ **Wh per request**, corresponding to a $100\times$ energy reduction relative to the naive baseline (LLaMA 8B-BF16 using the standard `transformers` backend) - achieved purely via improved batch consistency and GPU utilization.

## 5.3 LLaMA 70B: Scaling Benefits Hold at Large Scale

We repeated the experiment on LLaMA 3.1–70B ($4\times$H100s), keeping the same generation settings. Despite the $10\times$ increase in model size and the multi-GPU context, TGI achieved a per-request energy consumption as low as $2.4 \times 10^{-2}$ **Wh** - significantly lower than the naive baseline for 8B-BF16 ($1.2 \times 10^{-1}$ **Wh**). This confirms that dynamic batching and traffic shaping scale effectively to large models and hardware setups (Figure 3b).

## 5.4 Interpretation and Mechanisms

Two main mechanisms explain TGI's strong performance:

- **Continuous batching**: Incoming requests are incrementally batched at the token level as they arrive. Feedforward operations (e.g., MLP, QKV projections) are executed jointly across all active sequences, while attention is batched via paged mechanisms that group memory accesses efficiently across requests. This allows dynamic, low-latency batching without waiting for full prompts.

- **Kernel fusion and caching**: Fused operations (e.g., QKV projections, FFN layers) reduce intermediate memory writes and improve cache locality, further lowering DRAM usage and power draw.

Arrival shaping directly affects both mechanisms. Regular spacing ensures a steady stream of aligned requests, minimizing idle GPU time and improving the average batch size. Random delays still help by introducing jitter, but fixed spacing offers the most consistent utilization.

**Summary.** TGI combines efficient kernel execution with continuous batching strategies that adapt to incoming traffic. By shaping request arrivals - even with lightweight delay patterns - one can drastically improve batching quality and reduce energy consumption. These results suggest that

**user-side scheduling and backend inference optimizations are jointly critical** to making LLM deployment more sustainable.

## 6 MACRO IMPACT ESTIMATE

To contextualize our results, we estimate the energy footprint of serving the LLaMA 8B model at scale. In our baseline setup (`float32`, no batching), the mean GPU energy per request is $1.2 \times 10^{-1}$ **Wh** (Figure 3a). At $10^6$ requests per day, this yields:

$$\text{Total\_energy} = 10^6 \times 1.2 \times 10^{-1} \text{ Wh} = 1.2 \times 10^2 \text{ kWh/day}$$

This is equivalent to the daily electricity use of **over 10 French households**[2].

With optimized serving - using `bfloat16`, TGI, and regular arrival intervals - the mean energy drops to $1.1 \times 10^{-3}$ **Wh/request**, yielding:

$$\text{Total\_energy} = 10^6 \times 1.1 \times 10^{-3} \text{ Wh} = 1.1 \times 10^0 \text{ kWh/day}$$

This corresponds to a $> 100\times$ **reduction**, achieved solely through system-level improvements. These results emphasize that sustainable LLM deployment depends not only on model size or architecture, but also on scheduling and infrastructure.

## 7 RELATED WORK

**Environmental Impact of Inference.** While early works on AI sustainability focused on training (Luccioni et al., 2022; Strubell et al., 2019; Schwartz et al., 2019; Henderson et al., 2020; Patterson et al., 2021), inference has recently drawn attention due to its increasing share in real-world deployments (Wu et al., 2021; Luccioni et al., 2024a). Studies have quantified the energy per query (Samsi et al., 2023), compared hardware efficiency across CPUs and GPUs (Everman et al., 2023), highlighted the role of prompt length and verbosity (Gao et al., 2024; Poddar et al., 2025; Wilkins et al., 2024a), advocated for standardized reporting (Luccioni et al., 2024b; Tschand et al., 2025) and improved cost indicators (Dehghani et al., 2021). However, most focus on static benchmarks; few (Fernandez et al., 2025) address dynamic settings or system-level optimizations.

**Quantization and Precision.** Low-precision formats (`float16`, `bfloat16`, `int8`, `int4`) reduce memory and compute costs via techniques such as weight-only quantization (Dettmers et al., 2022; Frantar et al., 2023; Dettmers et al., 2023), activation-aware quantization (Lin et al., 2024), FP8 (Micikevicius et al., 2022), or post-training smoothing (Xiao et al., 2024). While some studies address energy impacts (Rajput & Sharma, 2024; Husom et al., 2025), real-world gains can vanish due to memory bottlenecks, dequantization overheads, or poor scaling (Lin et al., 2025). Energy-accuracy trade-offs are explored (Moons et al., 2017), but most analyses lack kernel- or phase-level granularity.

**Batching and Padding.** Batching improves throughput by amortizing overheads, but can introduce padding inefficiencies (Liu et al., 2024). The effectiveness depends on phase characteristics: decode benefits from batching due to shared memory access, while prefill may suffer from variable sequence lengths (Fernandez et al., 2025; Wilkins et al., 2024b; Patel et al., 2024). Dynamic batch shaping strategies (Agrawal et al., 2023; Spector & Re, 2023) are often necessary.

**Serving Infrastructure and Scheduling.** Modern inference engines like TGI (Face, 2022) and vLLM (Kwon et al., 2023) implement continuous batching (Yu et al., 2022), kernel fusion (Dao, 2023; Hsu et al., 2025), and paged attention (Kwon et al., 2023), greatly improving utilization. TensorRT-LLM and Triton Inference Server (NVIDIA, 2023; 2019) offer complementary low- and high-level optimizations for efficient LLM inference. Scheduling techniques such as query routing (Ding et al., 2024) and speculative decoding (Leviathan et al., 2023) further optimize latency and throughput.

---

[2]Based on an average of 4,255 kWh/year per household in France, i.e., ∼11.7 kWh/day. Source: `https://www.fournisseurs-electricite.com/compteur/consommation-electrique/moyenne`

However, these techniques can have mixed effects on energy (Shi et al., 2025), highlighting the need for joint energy-aware design.

**Energy measurement frameworks.** Tools such as CodeCarbon (Courty et al., 2024), pyRAPL (pyRAPL contributors, 2020), and NVIDIA's NVML enable reliable tracking of energy consumption during model execution.

**Summary.** While prior work provides building blocks - quantization, batching, dynamic serving- few studies jointly evaluate their impact on energy efficiency in real deployment conditions. We bridge this gap by dissecting LLM inference into phases and analyzing how system-level choices affect energy use across a wide operational range.

## 8 LIMITATIONS AND FUTURE WORK

While our analysis offers fine-grained insights into the energy and latency behavior of LLMs, several limitations remain:

**Prompt and output diversity.** Our experiments use relatively short prompts, keeping us within the linear scaling regime. Real-world usage may involve longer multi-turn dialogues or structured instructions, requiring non-linear models of compute cost and more diverse benchmarks.

**Transferability to other hardware.** Our results are based on NVIDIA H100 GPUs. While we expect qualitative trends (e.g., batching benefits, memory vs compute regimes) to hold, detailed power and latency behavior will differ on other accelerators (e.g., AMD, AWS Inferentia, TPU). Extending this analysis to other platforms is crucial for generalization.

**System-level effects.** Our energy measurements focus on GPU consumption only. CPU usage, memory transfers, and network I/O may contribute significantly to the system-level footprint, especially in multi-GPU or multi-node setups. Future work could account for these factors to provide a holistic view of inference efficiency.

## 9 CONCLUSION AND TAKEAWAYS

Energy efficiency in LLM inference is not solely dictated by model architecture or size. Instead, our experiments reveal a complex interplay between numerical precision, batch shaping, and serving configuration - each of which can dramatically affect latency and power draw.

- **Precision matters - but only in compute-bound regimes.** Lower-precision formats (e.g., `bfloat16`, `int8`) yield significant speedups and energy savings during prefill, particularly for large models. However, in memory-bound phases like decoding, quantization often fails to improve - and may even worsen - efficiency due to overheads like dequantization.
- **Batching is critical to efficiency.** Both static and dynamic batching reduce energy per token by improving hardware utilization and amortizing overheads. However, prefill is sensitive to padding inefficiencies, requiring careful shaping (e.g., bucketing) to avoid regressions.
- **Serving infrastructure shapes sustainability.** Our experiments with TGI demonstrate that the *how* of inference - i.e., the scheduling of requests - can impact energy consumption by up to two orders of magnitude, even with the same model and hardware.
- **Energy profiling should be phase-aware.** Decode and prefill exhibit fundamentally different compute characteristics, and should be measured and optimized separately. Reporting aggregate energy alone may obscure key bottlenecks or inefficiencies.

Taken together, our findings argue for a more holistic view of inference efficiency - one that includes not just model optimization, but also system design and traffic shaping. As LLMs continue to scale and proliferate, such systemic improvements will be critical to making their deployment environmentally sustainable.

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

## A    LATENCY COMPARISON BY NUMERICAL PRECISION

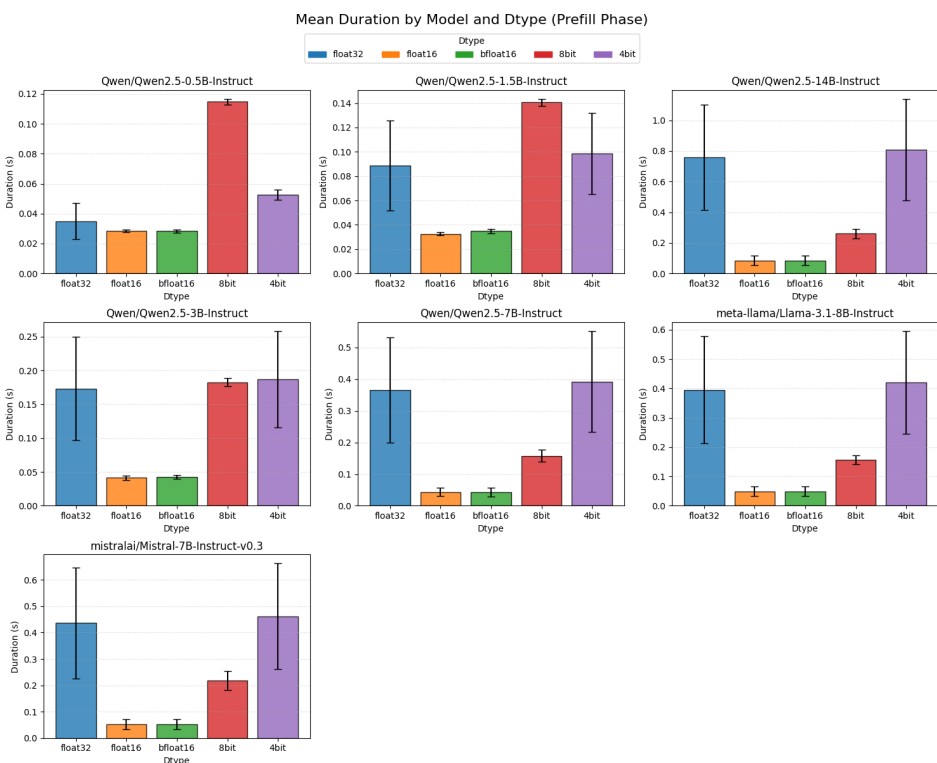

Figure 4: Mean latency per request (with variance across runs) for different models and data types during the **prefill** phase. Lower-precision formats generally reduce latency, with diminishing returns for already small models.

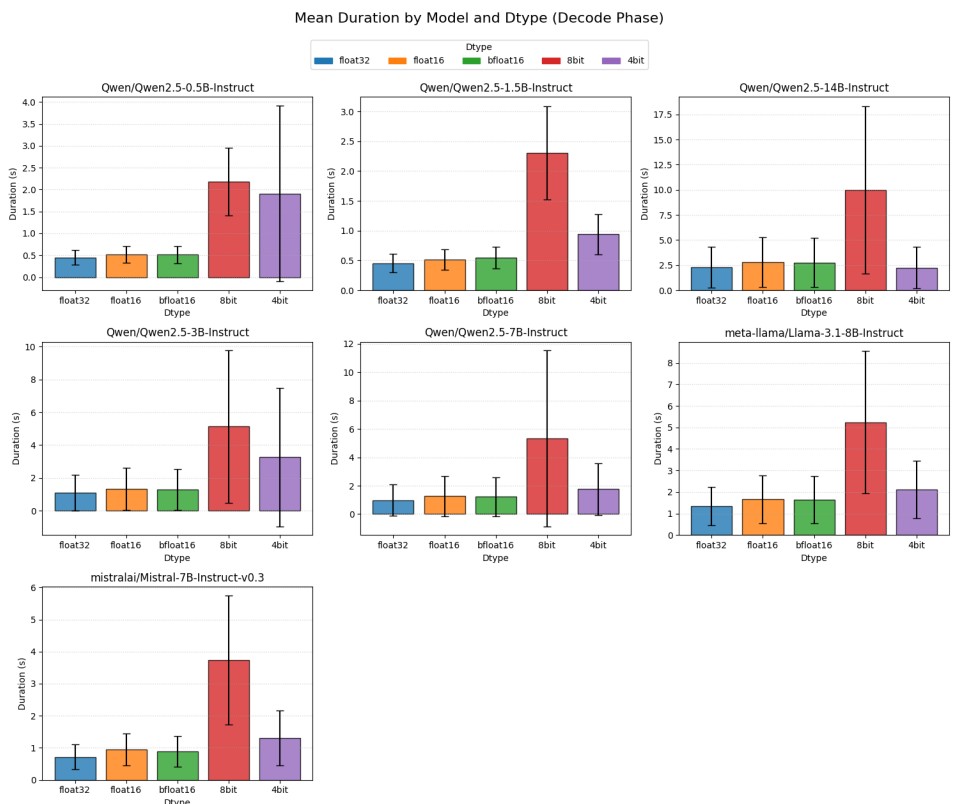

Figure 5: Mean latency per **generated token** (with variance across runs) for different models and data types during the **decode** phase. Memory-bound regimes lead to latency plateaus despite lower precision.

# B  LATENCY COMPARISON BY BATCH SIZE

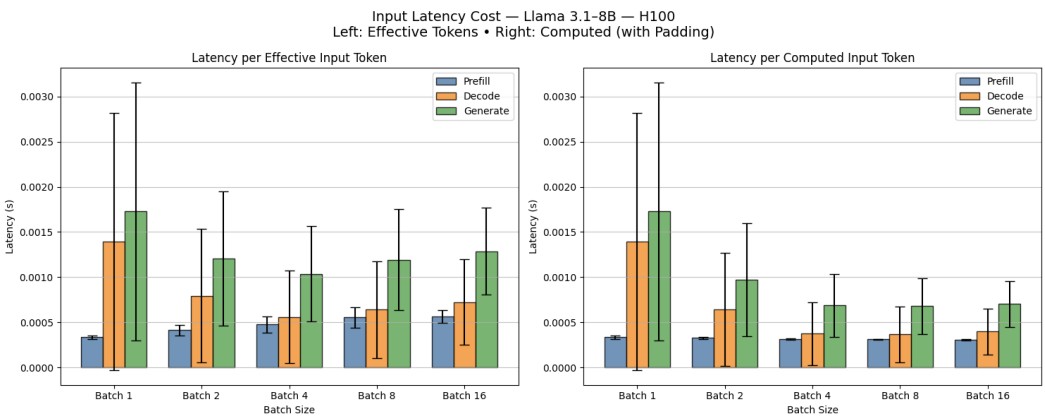

Figure 6: Latency per input token on LLaMA 3.1 8B across batch sizes. **Left:** Effective tokens only (excluding padding). **Right:** Computed tokens including padding overhead. Increasing batch size improves compute amortization but introduces padding-induced inefficiencies.

756
757
758
759
760
761
762
763
764
765
766
767
768
769
770
771
772
773
774
775
776
777
778
779
780
781
782
783
784
785
786
787
788
789
790
791
792
793
794
795
796
797
798
799
800
801
802
803
804
805
806
807
808
809

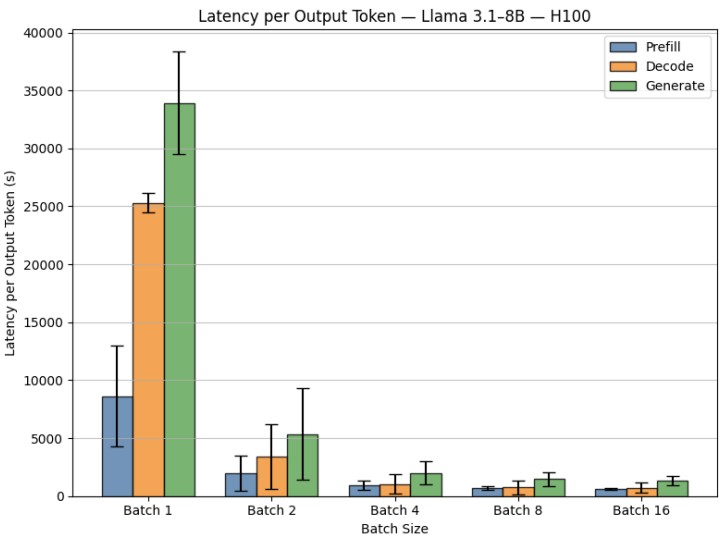

Figure 7: Latency per output token (effective = computed) across batch sizes. Gains plateau at moderate batch sizes due to limits in parallelism and autoregressive nature of decoding.

