# OpenReview forum: "Understanding Efficiency: Quantization, Batching, and Serving Strategies in LLM Energy Use"
_ICLR.cc/2026/Conference — Submitted to ICLR 2026_

### Official Review · Reviewer_4k4a · 2025-10-25

**Soundness:** 3
**Presentation:** 3
**Contribution:** 3
**Rating:** 6
**Confidence:** 4

**Summary:**

This paper presents a comprehensive empirical review of how system-level factors including quantization, batch size and serving strategies impact energy efficiency of LLMs on H100 GPUs across prefill and decode stages. Authors report that quantization only improves energy efficiency in the compute-bound stage (prefill) and not in memory-bound. They also observe that batching leads to energy efficiency up to an optimal batch size. Overall, they note that sustainable llm deployment requires phase-aware energy profiling and system-level optimization.

**Strengths:**

- Novel framework focusing on end-to-end serving stack instead of model or prompt-level energy measurements.
- Authors report results across multiple models and five numerical precisions. Experiments are carefully controlled and code is shared.
- The discussion around memory-bound vs compute-bound transitions, Tensor Core activation, and dequantization overheads demonstrates solid understanding of GPU behavior.
- Testing Hugging Face TGI with varied inter-arrival patterns connects lab results to production relevance. Showing that scheduling alone yields 100× savings is impactful.

**Weaknesses:**

- Experiments are confined to one single GPU generation. It would be great if these observations can be extrapolated to other hardware setups, including more recent GPU generation.
- The work remains largely empirical. It would be better to include a simple analytical model providing roofline-style analysis.
- CPU, DRAM, and network energy are estimated heuristically via CodeCarbon rather than directly measured, omitting parts of the total system energy footprint.
- The paper does not discuss energy–accuracy trade-offs at lower precisions, which would contextualize the value of aggressive quantization.
- Figures omit standard deviation or confidence intervals, which are important for understanding measurement variance.

**Questions:**

See weaknesses

---

> ### Author Response · Authors · 2025-11-20
> **Response to Reviewer 4k4a**
>
> We thank the reviewer for the positive assessment of our framework, the clarity of our experimental setup, and the recognition of the relevance of phase-aware decomposition and serving-stack analysis. We address each point below.
>
> ## 1. Hardware generalization beyond a single GPU generation
> We agree that expanding to other accelerators would enrich the study. Our primary goal, however, is to identify mechanisms - compute- vs memory-bound regimes, padding inflation, dequantization overheads, arrival-shaping effects - that are architecturally grounded and not specific to H100.
> While our main microbenchmarks use a single H100 SXM GPU, our serving-stack experiments include a 4×H100 setup for LLaMA 70B, showing that the same scheduling and dynamic batching effects persist at larger scale and in multi-GPU configurations. In the revised version, we will emphasize this broader setting and clarify how the identified mechanisms are expected to transfer across architectures with different FLOP/W and memory bandwidth characteristics.
>
> ## 2. CPU, DRAM, and network energy measurement
> We appreciate the reviewer’s concern. To clarify:
> - CPU energy is not estimated heuristically. We rely on pyRAPL, which uses RAPL counters on the AMD EPYC host to measure CPU energy directly.
> - RAM energy is estimated heuristically by CodeCarbon (as acknowledged), because hardware-level DRAM sensors are not exposed on our platform.
> - Network energy is negligible in our controlled local setup (no NIC activity beyond localhost), but we agree this would matter in distributed serving.
> We will make these distinctions explicit to avoid the impression that all non-GPU measurements are heuristic.
>
> ## 3. Lack of a simple analytical model
> We agree that an analytical perspective, such as a roofline-style view, can help contextualize empirical observations. Our focus in this work is the mechanistic explanation of phase behavior based on measured kernel-level effects rather than the development of a formal model. While we do not extend the paper with new analytical derivations, we will clarify in the camera-ready how the observed transitions between compute-bound and memory-bound regimes can be interpreted through standard concepts such as arithmetic intensity, memory transaction granularity, kernel fragmentation, and idle-time behavior. This addition stays within the scope of the current results and does not require additional experiments.
>
> ## 4. Omission of accuracy–energy trade-offs
> Our focus is on deployment-level energy mechanisms (quantization overheads, batch shaping, scheduling) that do not affect accuracy. That said, we agree that discussing accuracy implications of aggressive quantization would provide context. We will add a brief note explaining that:
> - weight-only int8/int4 used in our study preserves accuracy in typical inference workloads,
> - but more aggressive formats (e.g., 2–4 bits with activation quantization) introduce accuracy–efficiency trade-offs that are orthogonal to the system-level behaviors we examine.
>
> ## 5. Missing variance (std/confidence intervals) in figures
> We appreciate the reviewer’s concern. To clarify: variance is already displayed in all microbenchmark figures through error bars representing the standard deviation across repeated runs (10 trials per configuration after warmup). These plots exhibit very low variance due to the isolated setup.
> The only exception is the TGI experiments, where we do not show per-sample variance. This is not an omission but a limitation of the measurement setting: TGI batches and schedules requests dynamically, making it difficult to attribute energy fluctuations to individual samples in a granular way. As a result, reporting variance at the per-request level is not meaningful in this context.
> We will clarify this point in the revised manuscript to avoid confusion.
>
>
> ## Final remarks
> We thank the reviewer again for the constructive feedback. The requested points mainly concern clarification rather than additional experimentation, and we will update the manuscript accordingly. We believe these revisions will strengthen the presentation while preserving the focus on phase-aware, system-level insights for energy-efficient LLM inference.

---

### Official Review · Reviewer_P7xL · 2025-10-31

**Soundness:** 2
**Presentation:** 2
**Contribution:** 1
**Rating:** 2
**Confidence:** 4

**Summary:**

This paper empirically studies the energy and latency efficiency of LLM inference, examining how numerical precision, batching, and request scheduling affect performance. The authors benchmark several open-source models (Qwen, Mistral, LLaMA) across five precisions on NVIDIA H100 GPUs, measuring energy via CodeCarbon and separating prefill and decode phases. While the experiments are thorough and provide useful system-level insights, the work is largely descriptive with limited novelty or analysis, offering practical observations rather than new methodological contributions.

**Strengths:**

1. This paper addresses an important problem—the efficiency and energy consumption of LLM inference—which has become increasingly critical as model sizes continue to grow.
2. The authors examine this issue from multiple perspectives, including **quantization**, **batching**, and **serving strategies**, all of which are highly relevant for practical deployment and provide valuable insights for practitioners seeking to better understand and optimize energy use.

**Weaknesses:**

1. The contributions of this work are rather limited, as most of the findings and conclusions are already known in prior literature.
2. The experimental scope is not comprehensive. While some limitations—such as prompt and output diversity and hardware variety—are acknowledged, these aspects are crucial for a complete understanding of inference efficiency and should not be deferred. Moreover, key factors such as evaluations on larger models, studies on Mixture-of-Experts (MoE) architectures, and comparisons across different inference backends (vllms, sglang, etc.) and serving frameworks are missing.
3. The paper lacks an analysis of the trade-offs between task accuracy and deployment efficiency, which would be valuable for guiding practical decisions.
4. The overall presentation could benefit from a more formal and structured scientific writing style.

**Questions:**

1. How do 2-bit, 5-bit, and 6-bit quantization affect energy consumption, and how do quantization methods such as GPTQ and AWQ influence those results?
2. What is the impact of FP8 precision on energy efficiency compared to lower-bit integer formats and FP16/BF16?
3. How does LLM pruning (structured and unstructured) affect inference energy use, and what are the trade-offs between pruning ratio, latency, and overall energy savings?
4. How well do the observed energy and latency patterns generalize across different GPU architectures (e.g., A100, H100, H200) given differences in memory bandwidth, tensor core design, and power efficiency?

---

> ### Author Response · Authors · 2025-11-20
> **Response to Reviewer P7xL**
>
> We thank the reviewer for the thoughtful feedback. Below we address the main concerns and clarify the contribution.
>
> ## 1. “Limited novelty / descriptive analysis”
>
> We respectfully disagree. Prior work reports aggregate energy, but does not *decompose* inference into prefill vs decode or analyze how precision, batching, padding, kernel fusion, and arrival shaping affect these phases differently.
> Our contribution is this **phase-aware mechanism analysis**, which:
>
> * explains why the same technique helps in one phase but hurts in another,
> * reconciles inconsistent findings across the literature,
> * connects kernel-level behavior (fusion, fragmentation, KV-cache access, idle time) to energy trends.
>   We will emphasize this conceptual framework more clearly in the paper.
>
> ## 2. “Limited experimental scope”
>
> Our goal was to isolate *general mechanisms*, not benchmark every model or engine.
> The phenomena we analyze - compute- vs memory-bound behavior, dequantization overhead, padding inflation, batch-size optima, and arrival shaping - apply broadly (MoE, SGLang, TensorRT-LLM, larger models).
> We cannot expand experiments for the camera-ready, but will discuss how these mechanisms transfer to MoE (e.g., routing affects batch composition) and how different serving engines may shift magnitudes without altering the underlying trends.
>
> ## 3. “Missing accuracy–efficiency analysis”
>
> Most techniques discussed (batch shaping, scheduling, fusion) do **not** alter model weights, so accuracy is unaffected.
> Accuracy–efficiency trade-offs (quantization, pruning) are orthogonal to our core contribution; we will make this explicit.
>
> ## Reviewer Questions
>
> ### Q1. 2–6 bit quantization; GPTQ vs AWQ
>
> Ultra-low-bit formats magnify the effects observed for int4: heavier dequantization, irregular memory access, and kernel fragmentation.
> Thus decode energy often *increases* unless using fully fused kernels (e.g., QServe).
> GPTQ vs AWQ changes accuracy, not dequantization cost; energy is dominated by kernel structure.
>
> ### Q2. FP8
>
> FP8 reduces memory more than FP16/BF16 and keeps Tensor Core acceleration.
>
> * Prefill (compute-bound): slightly lower energy than BF16.
> * Decode (memory-bound): similar to FP16/BF16, as memory bandwidth dominates.
>   FP8 does not eliminate the core bottleneck.
>
> ### Q3. Pruning
>
> Structured pruning reduces both FLOPs and memory traffic → real energy savings, especially in prefill.
> Unstructured pruning rarely reduces memory traffic and can increase latency due to irregular sparsity.
> Our phase decomposition predicts which pruning settings meaningfully reduce energy.
>
> ### Q4. Portability (A100, H100, H200)
>
> Magnitudes vary, but regimes are consistent:
>
> * A100 becomes compute-bound later.
> * H200 raises bandwidth ceilings but decode remains KV-cache–dominated.
> * Tensor Core differences mostly affect prefill.
>   We will add a brief discussion of this expected portability.
>
> ## Final remarks
>
> While we cannot extend experiments, we will strengthen the paper by:
>
> * clarifying the conceptual contribution,
> * discussing scope limitations,
> * situating our findings relative to quantization/pruning/FP8/MoE literature.
>
> We hope this addresses the reviewer’s concerns.

---

### Official Review · Reviewer_Kmpb · 2025-11-01

**Soundness:** 3
**Presentation:** 3
**Contribution:** 2
**Rating:** 4
**Confidence:** 4

**Summary:**

This submission studies the GPU energy consumption impact of numerous systems-level design choices, such as adopted quantisation scheme, batching strategy and request scheduling, during LLM (inference) serving on H100 GPUs, across a diverse set of small-medium sized models.

**Strengths:**

- This submission conducts a timely and interesting analysis, on the increasingly important topic of quantifying the energy footprint of AI deployment, through the unique angle of incorporating system-level design aspects in the equation.
-The results are presented with appropriate breakdowns and normalisations to allow drawing clear conclusions in the examined setup.
- Many of the presented findings are insightful (with some being more expected than others), and the results presented can act as a point of reference for future work, as well as deployment practices, revealing the energy impact of usually underestimated factors such as padding and dequantisation operations.

**Weaknesses:**

- The main drawback of the presented methodology is the fact that energy consumption analysis is solely focused on GPU operations, ignoring memory transfers to/from GPU memory. This aspect often has decisive impact in the power draw, as well as is notably affected by some of the examined angles such as quantisation. As such, the practical applicability of the reported findings is limited, since the analysis does not seem to capture the whole picture.
- The presented analysis (e.g. on batching) could be strengthened if the conducted experiments and results included greater variability and breakdown in input/output length combinations, to be representative of different usecases (e.g. chat, translation and summarization demonstrate fundamentally different patterns in this aspect).
- The presented results would be more conclusive about the key underlying trade-off between energy and latency under different system-design choices, if the latency results from the Appendix were jointly presented (e.g. overlayed) and most-importantly commented in the respective sections of the analysis.

**Questions:**

Please consider replying in the comments raised in the weaknesses section above.

---

> ### Author Response · Authors · 2025-11-20
> **Response to Reviewer Kmpb**
>
> We thank the reviewer for the constructive feedback, the recognition of our methodological clarity, and the appreciation of the insights our phase-aware analysis brings to understanding inference energy. We address the concerns below.
>
> ## 1. GPU-only energy measurement and memory-transfer visibility
> We acknowledge that our measurements focus on GPU power as reported through NVML, without separately instrumenting PCIe/NVLink memory transfers. This is a limitation of our setup.
> Clarification:
>  Our goal was to study within-GPU energy behavior under different regimes (compute-bound vs memory-bound) rather than to estimate the full system-level footprint. The mechanisms we highlight - padding inflation, dequantization kernel overhead, batch shaping, and idle-power dominance - are all primarily internal to GPU execution. We will clarify this scope more explicitly in the revised version.
> While we cannot extend the experimental setup at this stage - e.g., using board-level instrumentation or external power meters - we will emphasize in the camera-ready that our findings describe GPU-resident effects and that additional system-level factors (host memory bandwidth, PCIe saturation, NUMA placement) can further amplify or attenuate the trends we observe.
>
> ## 2. Coverage of input/output length variability
> We agree that input/output lengths strongly affect phase balance (prefill vs decode), padding cost, KV-cache behavior, and batching dynamics. In the submitted version, we restricted the analysis to a controlled length window to ensure comparability across models, precisions, and serving stacks.
> Although we cannot add additional experimental configurations for the camera-ready, we will add a discussion section describing how different usage patterns - chat (short prompt, short output), translation (long input, long output), summarization (long input, short output) - would shift:
> - compute vs memory dominance,
> - padding inflation under static batching,
> - the batch-size optimum in decode,
> - the relative sensitivity to quantization overheads.
> This clarification will make the generalization boundaries of our findings more explicit.
>
> ## 3. Joint analysis of energy and latency in the main text
> We appreciate the suggestion. Our rationale for placing latency results in the appendix was to avoid overloading the figures in the main body, but we agree that explicitly connecting energy and latency would strengthen the exposition.
> As we cannot restructure the figures at this stage, we will revise the text to incorporate the key latency trends from the appendix directly into the narrative, including:
> - why lower precision reduces latency in compute-bound prefill but does not reduce energy proportionally;
> - why decode latency plateaus in memory-bound regimes despite lower precision;
> - why batching lowers both latency and energy up to b≈4, after which padding and KV-cache cost dominate.
> This will give readers a more integrated understanding of latency–energy trade-offs even if the plots remain separate.
>
> ## Final remarks
> We thank the reviewer for highlighting these important aspects. While the scope of the camera-ready prevents us from adding further experiments, we will ensure that:
> - the GPU-centric nature of our measurements is clearly stated;
> - the limitations regarding length variability are explicitly discussed;
> - latency–energy trade-offs are more clearly articulated in the relevant sections.
> We believe these clarifications will strengthen the paper and address the reviewer’s concerns while preserving the core contributions of our study.

---

### Official Review · Reviewer_1ird · 2025-11-06

**Soundness:** 4
**Presentation:** 4
**Contribution:** 4
**Rating:** 8
**Confidence:** 4

**Summary:**

This paper investigates where LLM inference energy is actually spent by decomposing generation into two phases, prefill and decode, and measuring energy for each across multiple open-weight families on H100 GPUs, including LLaMA, Qwen, and Mistral at different parameter scales. Prefill at the studied context lengths is typically compute-bound, while decode is often memory-bound. This split explains why the same knob can help in one phase and hurt in the other. Lower numeric precision helps mainly in compute-bound prefill, where Tensor Cores improve throughput and reduce energy. In memory-bound decode, weight-only int8 or int4 can increase energy due to extra small dequantization kernels and limited bandwidth gains. For batching, the paper reports energy per token under two normalizations: for input tokens, padding inflates prefill energy as batch size grows; for output tokens, energy falls quickly with batch across phases, roughly logarithmically, since completed sequences drop from the batch and longer kernels amortize overheads. Decode shows a U-shaped curve with an optimum around b=4 in their setup. Finally, the serving stack and traffic timing matter a lot: switching from a naive Transformers server to Hugging Face TGI with continuous batching cuts mean energy per request by about 12.5x, and adding fixed inter-arrival spacing (for example, 500 ms) yields reductions near 100x; similar trends appear at larger scales and on multi-GPU runs.

**Strengths:**

The paper tells a clear story: if you want to cut inference energy for LLMs, you must reason by phase. By decomposing generation into prefill and decode and normalizing energy by input and output tokens, the authors reveal that these phases live in different regimes (compute vs memory). This framing is simple yet novel, and it immediately explains why a single knob can help in one phase and hurt in the other. It also shifts the conversation from model-only tweaks to the full serving stack, showing that scheduling and batching policies can dominate outcomes.

The evaluation stands out by explaining results through mechanisms rather than surface-level curves. Precision is tested where it matters: lower formats help prefill when Tensor Cores are well utilized, but in decode energy can increase with weight-only int8 or int4 due to dequantization overheads and limited bandwidth relief. The batching study is explicit about normalization choices: energy per input token reveals padding waste in prefill, while energy per output token shows gains from longer kernels and sequences dropping out of the batch. Beyond microbenchmarks, the paper evaluates realistic serving backends and traffic patterns, showing that continuous batching and simple arrival pacing can deliver order-of-magnitude energy savings without altering the model.

**Weaknesses:**

1. Serving-stack breadth is limited. Results highlight TGI, but there is no controlled comparison vs vLLM or TensorRT-LLM under the same traffic.

2. System scope is GPU-centric, not resource-limited. The study focuses on high-end GPUs (H100) and GPU energy, with little accounting for CPU, or constrained devices.

3. Sequence-length coverage. Sequence length directly shapes which phase dominates (prefill vs decode), how much padding waste you incur, how large and active the KV-cache becomes, and how batching/continuous batching behave. Longer prompts stress prefill compute; longer outputs shift the workload toward memory traffic and cache updates; multi-turn chat effectively increases both over time. The study could benefit from including this in the experiments.

**Questions:**

- For weaknesses 1 and 2: while the core insights likely transfer across serving stacks and hardware, absent a controlled comparison the extent of portability is unclear. Reproducing key results with the same methodology (identical traffic traces, metrics, and normalizations) on vLLM and TensorRT-LLM, and rerunning a minimal subset on A100 or L40S as well as a CPU-heavier configuration, would materially strengthen the claims.

- Regarding weakness 3 (sequence-length coverage): the impact of prompt/output length on phase balance, padding waste, KV-cache growth, and batching behavior is central to your claims, so targeted length-controlled experiments would strengthen external validity. Something like bucket results by prompt/output length and reporting phase energy and batch optima would go a long way to improve the study.

---

> ### Author Response · Authors · 2025-11-20
> **Response to Reviewer 1ird**
>
> We thank the reviewer for the clear summary, the positive assessment of soundness and contribution, and the helpful suggestions. We address the comments below.
>
> ## Serving-stack breadth (vLLM, TensorRT-LLM)
> We agree that a controlled comparison across serving engines would provide additional external validity. Our study focused on TGI because it offered the instrumentation required for phase-level decomposition and traffic-controlled experiments. While we cannot extend the experimental scope in the current submission, we note that the mechanisms highlighted in our analysis (compute/memory-bound regimes, dequantization overheads, padding inflation, and arrival-shaping effects) depend primarily on kernel behavior rather than on backend-specific optimizations. We will clarify this point in the revised text.
>
> ## Hardware scope (GPU-centric, H100)
> We acknowledge the limitation of evaluating only on H100 and mainly reporting GPU energy. Our goal was to characterize regimes and phase behavior, which transfer across accelerators even though absolute magnitudes differ. We will add a discussion section explaining this more explicitly and outlining how CPU and heterogeneous setups might shift bottlenecks.
>
> ## Sequence-length coverage
> We agree that prompt and output lengths are a central factor in shaping prefill/decode balance, padding waste, and batch dynamics. Due to space and budget constraints, we restricted the main experiments to a controlled length band. We will add a concise discussion noting how length affects (i) compute vs memory bounds, (ii) padding sensitivity in prefill, and (iii) KV-cache pressure in decode, and we will clarify the expected trends for longer contexts.
>
> ## Answers to the reviewer’s questions
> ### - Portability across serving stacks and hardware
> We concur that a full controlled comparison would strengthen the claims. While we are unable to provide additional experiments at this stage, we will make explicit in the paper that our conclusions focus on mechanisms (e.g., continuous batching improving amortization; quantization interacting with memory-bound decode) and are expected to generalize, though backend-specific optimizations may change the magnitude of the gains.
> ### - Sequence-length experiments
> As noted above, we cannot add new experiments, but we will include a brief discussion explaining the expected effect of longer prompts/outputs on phase distribution and batch-size optima, addressing the reviewer’s concern.
>
> ## Final remarks
> We thank the reviewer again for the detailed and insightful evaluation. We appreciate that the suggestions mainly concern external validity, and we will update the manuscript to make the study’s scope and generalization boundaries clearer. We believe these clarifications will address the reviewer’s concerns while preserving the core contributions of the work.

---

### Meta-Review · Area_Chair_pUnF · 2025-12-29

**Summary:**

* All reviewers noted reliance on a single GPU generation, and narrow prompt/output length settings, making it hard to assess generality of the conclusions.
* Multiple reviewers raised concerns that the analysis focuses primarily on GPU energy, omitting or only heuristically treating CPU, memory, and system-level effects that are central to real deployment energy costs.
* The rebuttal mainly provides clarifications and additional discussion, without new experiments, measurements, or analyses to close the identified gaps. As a result, it is unlikely to have addressed the negative reviewers’ concerns or changed their assessments.

**Reviewer Concerns:**

Reviewer 1ird
* [Addressed] Authors acknowledged the lack of controlled comparisons and the GPU-centric scope. They said they will clarify both in revised text.
* [Not addressed] No new experiments, no additional hardware, no added sequence-length sweep. No updated PDF was uploaded. The response only promises revisions.

Reviewer Kmpb
* [Addressed] Authors answered the GPU-only critique and promising to clarify scope in the revised version.
* [Not addressed] No new experiments, the concerns are largely not addressed.

Reviewer P7xL
* [Addressed] Authors disagreed with "limited novelty" and answer the reviewer’s specific questions at a conceptual level. They promise to emphasize contribution and add discussion to the camera-ready version of the paper.
* [Not addressed] No additional experiments, no MoE evaluation, no backend comparisons. Only intent to strengthen the paper via discussions. The concerns are largely not addressed.

Reviewer 4k4a
* [Addressed] Authors responded point-by-point and commit to clarifications in the revised/camera-ready manuscript.
* [Not addressed] No added non-H100 hardware experiments, no new analytical model beyond explanation, etc. No PDF updates, nothing explains an inability to update it.

While the authors repeatedly mention that revisions, additions, or clarifications will be made in the revised or camera-ready version, the PDF was not updated as part of the rebuttal, leaving it unclear why.

**Reviewer Scores:**

* Reviewer 1ird: Would have kept the same score --> 8 (from 8).
* Reviewer Kmpb: Would have kept the same score --> 4 (from 4). Methodological limitations are not resolved.
* Reviewer P7xL: Would have kept the same score --> 2 (from 2). Reviewer’s rejection is driven by lack of novelty and missing experiments, unchanged by rebuttal.
* Reviewer 4k4a: Would have kept the same score --> 6 (from 6).

---

### Decision · Program_Chairs · 2026-01-26

Reject